# Characterization of Silver Carbonate Nanoparticles Biosynthesized Using Marine *Actinobacteria* and Exploring of Their Antimicrobial and Antibiofilm Activity

**DOI:** 10.3390/md21100536

**Published:** 2023-10-13

**Authors:** Omar Messaoudi, Ibrahim Benamar, Ahmed Azizi, Salim Albukhaty, Yasmina Khane, Ghassan M. Sulaiman, Mounir M. Salem-Bekhit, Kaouthar Hamdi, Sirine Ghoummid, Abdelhalim Zoukel, Ilhem Messahli, Yacine Kerchich, Farouk Benaceur, Mohamed M. Salem, Mourad Bendahou

**Affiliations:** 1Department of Biology, Faculty of Science, University of Amar Telidji, Laghouat 03000, Algeria; o.messaoudi@lagh-univ.dz (O.M.); ib.benamar@lagh-univ.dz (I.B.); k.hamdi.bio@lagh-univ.dz (K.H.); goumidsirine@gmail.com (S.G.); messahli.ilhem@gmail.com (I.M.); f.benaceur@lagh-univ.dz (F.B.); 2Laboratory of Applied Microbiology in Food and Environment, Abou Bekr Belkaïd University, Tlemcen 13000, Algeria; bendahou63@yahoo.fr; 3Department of The Common Trunk Sciences and Technology, Faculty of Technology, University of Amar Telidji, Highway Ghardaia, P.O. Box G37 (M’kam), Laghouat 03000, Algeria; a.azizi@lagh-univ.dz; 4Department of Chemistry, College of Science, University of Misan, Maysan 62001, Iraq; 5College of Medicine, University of Warith Al-Anbiyaa, Karbala 56001, Iraq; 6Faculty of Science and Technology, University of Ghardaia, BP455, Ghardaia 47000, Algeria; yasminekhane@yahoo.fr; 7Division of Biotechnology, Department of Applied Sciences, University of Technology, Baghdad 10066, Iraq; ghassan.m.sulaiman@uotechnology.edu.iq; 8Department of Pharmaceutics, College of Pharmacy, King Saud University, P.O. Box 2457, Riyadh 11451, Saudi Arabia; mbekhet@ksu.edu.sa; 9Laboratory Physico-Chemistry of Materials, Laghouat University, Laghouat 03000, Algeria; abdelhalim.zoukel@gmail.com; 10Center for Scientific and Technical Research in Physicochemical Analysis (PTAPC-Laghouat-CRAPC), Laghouat 03000, Algeria; 11École Nationale Polytechnique (ENP), Laboratory of Environmental Science and Technology, El Harrach 16200, Algeria; yacine.kerchich@g.enp.edu.dz; 12Research Unit of Medicinal Plant (RUMP) Attached to Center of Biotechnology (CRBt, 3000, Constantine), Laghouat 03000, Algeria; 13College of Medicine, Huazhong University of Science and Technology, Wuhan 430030, China; medozcockney@gmail.com

**Keywords:** isolation, marine *Actinobacteria*, *Saccharopolyspora erythrea*, silver carbonate nanoparticles, antimicrobial, antibiofilm

## Abstract

Bacterial resistance to different antimicrobial agents is growing with alarming speed, especially when bacterial cells are living in biofilm. Hybrid nanoparticles, synthesized through the green method, hold promise as a potential solution to this challenge. In this study, 66 actinomycete strains were isolated from three distinct marine sources: marine sediment, the algae *Codium bursa*, and the marine sponge *Chondrosia reniformis*. From the entirety of the isolated strains, one strain, S26, identified as *Saccharopolyspora erythrea*, was selected based on its taxonomic position and significant antimicrobial activity. Using the biomass of the selected marine *Actinobacteria*, the green synthesis of eco-friendly silver carbonate nanoparticles (BioAg_2_CO_3_NPs) is reported for the first time in this pioneering study. The BioAg_2_CO_3_NPs were characterized using different spectroscopic and microscopic analyses; the synthesized BioAg_2_CO_3_NPs primarily exhibit a triangular shape, with an approximate size of 100 nm. Biological activity evaluation indicated that the BioAg_2_CO_3_NPs exhibited good antimicrobial activity against all tested microorganisms and were able to remove 58% of the biofilm formed by the *Klebsiella pneumoniae* kp6 strain.

## 1. Introduction

Bacterial resistance to different antimicrobial agents, such as antibiotics, disinfectants, antiseptics, and food preservatives, arises when microbial cells become no longer sensitive to the antimicrobial agents at the concentrations used in practice. This resistance has significant health, economic, and environmental implications [1]. Compared to planktonic cells, various studies have demonstrated that bacterial resistance increases significantly when cells are in the biofilm state. Biofilm represents a unique microbial lifestyle, where bacteria are attached to a solid surface and enclosed within a self-produced extracellular polymeric substance, primarily composed of exopolysaccharides and proteins [2,3]. Consequently, both natural and synthetic classes of antimicrobial agents are currently unable to evade this phenomenon [4]. The current progress in addressing both bacterial resistance and biofilm problems involves the application of metallic nanoparticles obtained from green synthesis. This cutting-edge solution, derived from nanobiotechnology, is considered a boon in both the medical and industrial domains [5,6,7]. Different metals have been used for nanoparticle biosynthesis; however, the recent trend in this field is the biosynthesis of hybrid nanoparticles, combining multiple metal or mineral components to form novel nanomaterials. This approach allows for the enhancement of their properties, offering promising potential to effectively combat microbial resistance and tackle biofilm-related issues. Different biological entities can be used for the biosynthesis of nanoparticles, such as plants, fungi, and bacteria. Among bacteria, actinomycetes are well known for being a promising source of bioactive metabolites, making them particularly suitable for the synthesis of eco-friendly, cost-effective, and non-toxic metallic nanoparticles [8,9,10,11].

These filamentous bacteria are widely distributed and can be isolated from various terrestrial and marine environments. One of the well-known marine environments recognized for its geo-biodiversity is the Mediterranean coastline, which hosts more than 7% of the world’s marine biodiversity, encompassing marine sediments, sponges, and algae, all of which are notable for their richness in actinomycete strains [12,13,14]. This coastal expanse holds the Algerian coast which spans 1622 km, and represents a yet untapped ecosystem, teeming a potential diversity of *Actinobacteria* strains, waiting to be discovered and harnessed [15,16,17]. Therefore, in response to the lack of published research material on such an important topic, this study sought to gather data on: (i) isolation, molecular identification, and assessment of the antimicrobial activity of marine actinomycetes isolated from sediments, algae, and marine sponges collected from the west coast of Algeria; (ii) the green synthesis of silver carbonate nanoparticles using marine *Actinobacteria* strain, followed by their characterization with a range of microscopic and spectroscopic techniques; and (iii) assessment of the biological activity of the biosynthesized silver carbonate nanoparticles.

## 2. Results and Discussion

### 2.1. Actinomycetes Isolation

In this study, we attempted to investigate the biodiversity of actinomycetes strains isolated from Algerian marine environments, such as sponges, algae, and marine sediment. The obtained results are summarized in Figure 1.

A total of 66 *Actinobacteria* strains were isolated from different marine sources collected from two sites along the Algerian coast, using four different culture media. The obtained *Actinobacteria* strains (66 isolates) were distributed based on their marine origins as follows: 46 strains were from marine sediment (70%), 15 isolates from the marine algae *Codium bursa* (23%), and 5 strains from the marine sponge *Chondrosia reniformis* (7%) (Figure 1A).

The selection of the two marine organisms, *Codium bursa* and *Chondrosia reniformis*, for actinomycete isolation in the current study was motivated by the absence of previous research on the diversity of *Actinobacteria* linked to these marine sources collected from the Mediterranean coast.

Previous studies have reported that sponges and algae host substantial microbial biomass, encompassing archaea, fungi, viruses, and bacteria. This symbiotic interaction yields diverse beneficial outcomes for both entities, including nutrient acquisition, metabolic processing, and chemical defense [18,19,20]. Moreover, several reports highlighted that marine sediments offer an excellent opportunity for the discovery of new microorganisms, which may contain a new gene cluster for the biosynthesis of novel natural products [21,22,23].

Furthermore, the Algerian coastal region stretches for 1622 km. Despite this long distance, this environment remains poorly investigated in terms of the biodiversity of microorganisms, including *Actinobacteria*, as an attractive source of new bioactive compounds, and only a few studies have been reported. Djinni et al., 2014 [24] isolated the marine strain *Streptomyces sundarbansensis* from brown algae collected from the Algerian coastline. This strain produces the [=2-hydroxy-5-((6-hydroxy-4-oxo-4H-pyran-2-yl)methyl)-2-propylchroman-4-one] compound, which exhibits activity against Methicillin-resistant *Staphylococcus aureus* (MRSA). In another study, Ouchene et al., 2021 [25] obtained numerous *Actinobacteria* strains from the coast of Bejaia City (Algeria), using different selective isolation methods.

Moreover, previous studies showed that the composition of culture media significantly affected the number and diversity of *Actinobacteria* strains [26,27,28]. Therefore, to enhance the isolation of *Actinobacteria* from the three marine sources, four culture media were used. The results indicated that SCA and M2 were the most efficient culture media for actinomycete isolation since 25 and 23 actinomycete strains were obtained from each culture medium, respectively. However, ISP_2_ and M4 provide only 10 and 8 *Actinobacteria* strains, respectively (Figure 1B).

### 2.2. Antimicrobial Activity of Actinobacteria Strains

*Actinobacetria* isolates were evaluated for their antibacterial activity against a panel of three Gram-negative bacteria and four Gram-positive bacteria, in addition to one yeast. The results are given in Appendix A. The results revealed that 40 strains out of 66 obtained isolates, showed antimicrobial activity at least against one of the tested microorganisms. From the active strains, 30 isolates were obtained from the marine sediment, while 7 strains were isolated from the marine algae *Codium bursa,* whereas three active strains were isolated from the marine sponge *Chondrosia reniformis.* Furthermore, 26 strains exhibited antifungal activity against the yeast *C. albicans*, while 29 strains were active against Gram-positive bacteria and 3 strains showed activity against Gram-negative bacteria; however, 4 strains exhibited activity against both Gram-positive and Gram-negative bacteria. Therefore, the *Actinobacteria* strains showed higher activity against Gram-positive bacteria than Gram-negative bacteria. This difference in sensitivity can be ascribed to the morphological difference between the cell walls of Gram-positive and Gram-negative bacteria, in particular, the presence of lipopolysaccharides in the outer membrane of Gram-negative bacteria, which makes the bacterial cell more impermeable to the lipophilic compounds. However, Gram-positive bacteria are more susceptible due to the absence of lipopolysaccharides [29]. Regarding resistance, *L. monocytogenes* was the most resistant bacteria tested. This resistance can be attributed to intrinsic mechanisms, such as efflux pumps, which can pump out or neutralize antimicrobial compounds, making the eradication of *L. monocytogenes* challenging [1]. In contrast, *Bacillus cereus* and *Micrococcus lotus* were the most sensitive tested microorganisms. The highest inhibition area values ranged from 21 to 30 mm, and they were observed for strain S26 against the four tested bacteria: *A. baumannii*, *S. aureus*, *M. luteus*, and *B. cereus*.

### 2.3. Biodiversity of Marine Actinobacteria

Partial 16S rRNA gene sequencing was used to identify 19 *Actinobacteria* strains that exhibit the highest antimicrobial activity against the tested microorganisms. By using BLAST analysis, the obtained sequences were compared to the closest homologous sequences. The results of molecular identification, shown in Appendix A, indicated that the identified isolates shared sequence similarities that ranged from 99 to 100% with 3 different *Actinobacteria* genera, including *Streptomyces* (11 strains), *Nocardiopsis* (7 strains), and *Saccharopolyspora* (1 strain). Previous studies indicated that the genera *Streptomyces*, *Nocardiopsis*, and *Saccharopolyspora* have been widely isolated from different marine sources, which were collected from distinct geographical locations, including marine sediments [30], marine sponges [31,32], marine invertebrates such as corals [33] and molluscs [34], and algae [16]. After the molecular identification of the 19 active strains, 9 actinobacterial isolates, which exhibited the best antimicrobial activity against the pathogenic microorganisms, were selected for phylogenetic study. This aimed to determine the phylogenetic relationships between the selected strains and the closest species. The results are shown in Figure 2. The obtained dendrogram differentiated the nine selected strains into three clusters. In fact, the strains V17, CS44, RH94, and S29, were spread in four different phyletic lines within the *Streptomyces* group, and they showed 99.50%, 99.30%, 99.75%, and 99.75% of sequence similarity, with type strains, *Streptomyces sampsonii* DSM 40394, *Streptomyces pseudogriseolus* strain NRRL B-3288, *Streptomyces fulvissimus* strain DSM 40593, and *Streptomyces armeniacus* DSM 43125, respectively. However, the isolates C19, CHB2, A58, and M23, which belong to the *Nocardiopsis* cluster (Figure 2), form three separated branches within the phylogenetic tree and are close to the species *Nocardiopsis terrae* YIM 90022 (99.30% similarity), *Nocardiopsis arvandica* DSM 45278 (100% similarity), and *Nocardiopsis dassonvillei* IMRU 509 (99.75% similarity), respectively. Strain S26 forms a separate clade, and it is close to the species *Saccharopolyspora erythraea*, with 99.70% similarity.

Considering the results of molecular identification coupled with antimicrobial activity, the strain S26 was selected for the preparation of silver carbonate nanoparticles (Ag_2_CO_3_NPs). This was because it belongs to a rare *Actinobacteria* genus, *Saccharopolyspora*, and because it was the most active isolate from the strains recovered from the marine Algerian environment (Appendix A).

### 2.4. Identification of the Selected Strain S26

The selected strain S26 was identified as belonging to the genus *Saccharopolyspora,* which includes 46 species with validly published names (LPSN: https://www.bacterio.net, accessed on 12 October 2023). The genus *Saccharopolyspora* is a member of the family *Pseudonocardiaceae*, order *Actinomycetales*, and phylum *Actinobacteria.* Molecular identification indicated that this strain was close to the species *Saccharopolyspora erythraea* DSM 40517^T^. Strains of this species are characterized by the formation of long spore chains that are surrounded by a spiny surface sheath. However, the other species belong to the *Saccharopolyspora* genus and form a smooth surface sheath [35]. Therefore, in order to verify whether the micromorphology of strain S26 is identical to the type strain *Saccharopolyspora erythraea* DSM 40517^T^ or not, scanning electronic microscopy was performed. The results (Figure 3) indicate that the strain S26 forms a branched substrate mycelium that is fragmented at maturity into coccoid and bacillary elements. The aerial mycelium of S26 forms a spore chain surrounded by a spiny sheath (Figure 3), and this micromorphology is typical of strains belonging to the species of *Saccharopolyspora erythraea* [35]. In addition, other similarities have been noted between the strain S26 and the species *Saccharopolyspora erythraea*, such as the use of different carbon sources by the isolate S26 being perfectly the same as that of the species *Saccharopolyspora erythraea*, except for the raffinose and the cellulose, which are not used by strain S26 (Appendix A). Furthermore, both strain S26 and the closest species, *Saccharopolyspora erythraea*, secreted a bright red soluble pigment, and they do not produce melanoid pigments (Appendix A). However, some macromorphological differences between S26 and *Saccharopolyspora erythraea* have been noted. In fact, the aerial mycelium of S26 is beige-brown, whereas it is usually white to pink for the species *Saccharopolyspora erythraea* [36]. More characteristics of strain S26 can be found in Appendix A.

On the basis of the genotypic and phenotypic data, we can conclude that the isolate S26 is identical to the species *Saccharopolyspora erythrea*. The 16S rRNA gene sequences of the strain S26, were deposited in GenBank under the accession number OQ567772.

### 2.5. Biosynthesis and Characterization of Silver Carbonate Nanoparticles

The present study aimed to describe a feasible and simple green method, involving the use of marine *Actinobacteria* for the biosynthesis of BioAg_2_CO_3_NPs, which can be used to overcome the microbial resistance toward antimicrobial agents.

#### 2.5.1. Biosynthesis of Silver Carbonate Nanoparticles

*Saccharopolyspora erythrea* S26 strain was selected from among 19 active marine actinomycete strains, for the biosynthesis of BioAg_2_CO_3_NPs because it showed potent antimicrobial activity against various tested microorganisms.

The biosynthesis of BioAg_2_CO_3_NPs by strain S26 has been primarily identified via color change; the results are shown in Appendix A.

The results indicated that, before the addition of the precursor, silver nitrate (AgNO_3_), the flask containing the biomass of strain S26 was off-white. Once the AgNO_3_ was added, the color changed to dark brown (Appendix A). This color change was the first indication of the formation of BioAg_2_CO_3_NPs by the biomass of strain *Saccharopolyspora erythrea* S26 [37]. These findings have been confirmed by DRX, XRF, and FTIR.

#### 2.5.2. Characterization of Silver Carbonate Nanoparticles

Different characterization techniques, such as Fourier Transform Infrared Spectroscopy (FTIR), X-ray fluorescence (XRF), X-ray diffraction (XRD), scanning electron microscopy (SEM), and atomic force microscopy (AFM), were employed to confirm the formation of silver carbonate nanoparticles by the biomass of strain S26 and analyze their properties.

#### 2.5.3. FTIR and XRF Analysis of Silver Carbonate Nanoparticles

FTIR and XRF analyses were conducted for the silver carbonate nanoparticles synthesized using the biomass of strain S26. The results are presented in Figure 4 and Table 1.

The FTIR spectrum is a valuable tool for identifying functional groups and chemical bonds. According to the FTIR spectrum presented in Figure 4, four characteristic absorption bands appeared at four different positions, namely 775.50 cm^−1^, 860.15 cm^−1^, 1162.59 cm^−1^, and 1464.27 cm^−1^, which are attributed to the carbonate groups (CO_3_^2−^) in BioAg_2_CO_3_NPs [38,39]. Therefore, the FTIR spectrum (Figure 4) conveniently demonstrates the presence of CO_3_^2−^, within the nanoparticles biosynthesized by the biomass of strain S26. In addition, the IR spectrum of BioAg_2_CO_3_NPs exhibited peaks at 3186.51 cm^−1^ and 2467.01 cm^−1^, corresponding to OH and N–H vibrations, respectively. These peaks indicate the presence of a surface-bound bio-organic phase that acts as a capping/stabilizing agent on the surface of the synthesized nanoparticles [40]. These organic agents have a significant impact on the chemical, physical, and biological features of the nanoparticles [41].

From the results presented in Table 1, which show the atomic content of the BioAg_2_CO_3_NPs, it is evident that the synthesized nanoparticles predominantly contain silver as a major component in their composition, which aligns with the expected composition of silver carbonate nanoparticles. Therefore, the presence of carbonate groups (CO_3_^2−^) and silver atoms in the composition of the synthesized nanoparticles, as indicated in Figure 4 and Table 1, respectively, offers valuable insights into the chemical makeup of the metallic particles, and suggests the successful synthesis of BioAg_2_CO_3_NPs, utilizing the biomass of strain S26. This compositional information has been further supported by XRD characterization of the synthesized nanoparticles.

#### 2.5.4. XRD Characterization of the Silver Carbonate Nanoparticles

The X-ray diffraction (XRD) analysis was performed in order to examine the crystallographic structure as well as to determine the size of BioAg_2_CO_3_NPs synthesized using the biomass of strain S26. The obtained results are shown in Figure 5.

The diffraction pattern of the BioAg_2_CO_3_NPs, as depicted in Figure 5, exhibits clearly defined peaks. The distinct diffraction peaks corresponding to the 2Ɵ values recorded in Appendix A, show well-matched positions and intensities when compared with the standard ICDD files no. 00-031-1237. This similarity strongly suggests that the sample is primarily made of silver carbonate nanoparticles in the hexagonal phase.

The identification of the silver carbonate nanoparticles in the hexagonal phase at high concentrations also raises questions regarding their size. A thorough investigation using the Williamson–Hall plot [42] (Figure 6) is warranted to gain insights into the crystallite size and lattice strain of the biosynthesized BioAg_2_ CO_3_NPs nanoparticles from X-ray diffraction data.

The strain was assumed to be uniform in all crystallographic directions. The plots showed a slight positive strain present in the BioAg_2_CO_3_NPs. This strain might be due to lattice expansion. Moreover, the extracted data from the plot clearly indicate that the average crystallite size of the BioAg_2_CO_3_NPs is 103 nm. A previous study conducted by Loncarevic et al., (2017) [43], reported the synthesis of rod-like Ag_2_CO_3_ nanoparticles in water through a precipitation reaction between NaHCO_3_ and AgNO_3,_ in the presence of polyvinylpyrrolidone, with a size range from 20 to 50 nm.

The synthesis of silver carbonate nanoparticles has been previously accomplished through a range of physical and chemical methods [43,44,45,46]. However, in comparison to these traditional synthesis techniques, the green synthesis of silver carbonate nanoparticles (BioAg_2_CO_3_NPs) stands out due to its simplicity, cost-effectiveness, non-toxicity, and environmentally friendly nature. Nevertheless, there is a lack of available literature on the green synthesis of silver carbonate nanoparticles so far, and this study can be regarded as the pioneer in reporting the green synthesis of this nanomaterial using a marine strain.

In addition, the inclusion of CO_3_^2−^ ions into silver carbonate nanoparticles enhances their biocompatibility, biodegradability, stability, chemical reactivity, and biological activity [47,48]. In fact, in comparison to numerous other inorganic materials, the presence of CO_3_^2−^ ions can facilitate nanoparticles’ interaction with biological systems, which can potentially reduce toxicity and enhance compatibility with living organisms, this renders them particularly suitable for medical applications [47,48,49].

#### 2.5.5. Microscopic Characterization of the Silver Carbonate Nanoparticles

Different microscopic methods, including scanning electron microscopy (SEM) and atomic force microscopy (AFM), were utilized to examine the morphology, size, and surface topology of the silver carbonate nanoparticles synthesized using the biomass of strain S26. The results are illustrated in Figure 7.

Microscopic analysis through SEM of silver carbonate nanoparticles biosynthesized using the biomass of strain S26 unveiled diverse morphological shapes and uniform distribution, with only a minor presence of particle aggregation (Figure 7A). The prevalent form observed was triangular (indicated by the green arrow) (Figure 7A), while other shapes, such as rectangular (indicated by the white arrow) and spherical, were also detected (Figure 7A). Moreover, the SEM results indicated that the average size of Bio Ag_2_CO_3_NPs ranges from 90 nm to 110 nm, thus confirming the findings from the X-ray diffraction data (Figure 6). Various parameters, including the preparation process, the strain used for synthesis, and the culture conditions such as pH and temperature, exert a significant influence on the morphology and size of the synthesized nanoparticles [50].

Atomic force microscopy (AFM) was used to provide a precise three-dimensional representation of the surface topology of the silver carbonate nanoparticles biosynthesized using the biomass of strain S26. The results indicate that the surface topology of the BioAg_2_CO_3_NPs displayed a triangular shape, as revealed by the AFM findings (Figure 7B). It is important to highlight that the geometrical shape directly affects various attributes, including surface area, optical properties, catalytic activity, as well as their interactions with surrounding environments, such as biological systems [51,52,53].

### 2.6. Biological Activity of Silver Carbonate Nanoparticles

#### 2.6.1. Antimicrobial Activity

The antibacterial activity of BioAg_2_CO_3_NPs was evaluated against a range of pathogenic bacteria and fungi using the agar well diffusion method. Furthermore, the minimum inhibitory concentration (MIC) against the same tested microorganisms was determined through the serial dilution method. The results are presented in Table 2.

Based on the results presented in Table 2, it is clear that the BioAg_2_CO_3_NPs synthesized by strain S26 exhibited notable antibacterial activity against the full range of examined pathogens, with inhibition diameters ranging from 11 mm to 21 mm (Appendix A). The MIC of nanoparticles synthesized using the biomass of strain S26 was subsequently assessed against each of the tested microorganisms, as detailed in Table 2. The results demonstrate that BioAg_2_CO_3_NPs are efficient antibacterial agents against all tested microbes, with MIC values ranging from 18.75 to 150 µg/mL. The biological activity of nanoparticles is significantly influenced by their characteristics. Several studies have shown that the size of nanoparticles and their antibacterial activity are inversely correlated; in other words, the smaller the nanoparticle size, the greater the surface interaction area with microbial cells, resulting in heightened antimicrobial activity. [54,55]. Additionally, the antimicrobial potential of BioAg_2_CO_3_NPs is also shape-dependent [56]. In this context, Pal et al. (2007) [57] found that triangular silver nanoparticles were more effective against *E. coli* compared to spherical nanoparticles.

The vulnerability observed in the tested bacteria and fungi towards BioAg_2_CO_3_NPs can be attributed to the nanoparticles’ ability to attack microbial cells simultaneously through multiple mechanisms, including protein inactivation, the production of reactive oxygen species (ROS) [58], and the formation of free radicals [59]. Consequently, pathogenic microorganisms become incapable of developing resistance to BioAg_2_CO_3_NPs, unlike traditional chemical antimicrobial agents, such as antibiotics and antifungals, which act on microbial cells via only one mechanism. Due to this rarity of microorganisms’ resistance to BioAg_2_CO_3_NPs, the use of these nanoparticles offers a potential alternative to overcoming multidrug-resistant microorganisms [60,61].

#### 2.6.2. Antibiofilm Activity

The microtiter dish biofilm formation assay was conducted to evaluate and assess the antibiofilm activity of BioAg_2_CO_3_NPs against the biofilm formed by *Klebsiella pneumoniae* kp6 strain. The experiment was performed in the presence of BioAg_2_CO_3_NPs, with a final concentration of 150 µg/mL, corresponding to the MIC determined against the kp6 strain (Table 2). The results can be found in Figure 8 and Appendix A.

The capacity of BioAg_2_CO_3_NPs to remove the biofilm formed over 24 h by *Klebsiella pneumoniae* kp6 strain was initially assessed through a visual comparison between the wells treated with BioAg_2_CO_3_NPs and those of the negative control (wells without BioAg_2_CO_3_NPs). A clear decrease in the color intensity of crystal violet was observed in wells exposed to BioAg_2_CO_3_NPs, suggesting good antibiofilm activity against the biofilm formed by the *Klebsiella pneumoniae* kp6 strain (Appendix A).

In order to quantify the anti-biofilm activity of BioAg_2_CO_3_NPs, UV-vis spectroscopy was performed. The results are presented in Figure 8.

As illustrated in Figure 8, the BioAg_2_CO_3_NPs synthesized using the biomass of strain S26 successfully eradicated 58% of the biofilm formed by the *Klebsiella pneumoniae* kp6 strain, confirming then the good antibiofilm activity of BioAg_2_CO_3_NPs, observed in Appendix A. Several authors have extensively investigated the efficacy of silver nanoparticles (AgNPs) in eliminating biofilms formed by various bacterial strains, however, to the best of our knowledge, this study was the pioneering investigation aimed to evaluate the antibiofilm activity of silver carbonate nanoparticles against the biofilm formed by *Klebsiella pneumoniae* kp6 strain.

The antibiofilm activity of BioAg_2_CO_3_NPs can be attributed to several mechanisms, including the inhibition of bacterial cell adhesion to the support surface [62], disruption of intermolecular forces [5], and inhibition of Quorum Sensing [63]. Additionally, BioAg_2_CO_3_NPs may potentially contribute to neutralizing EPS production, which represents the adhesive substance necessary for biofilm formation [64].

Furthermore, the size and shape of nanoparticles significantly influence their antibiofilm activity. Smaller nanoparticles exhibit higher antibiofilm activity, as their diminutive size allows them to penetrate biofilms and adhere to cell membranes, thereby interfering with permeability and metabolism [64,65]. Additionally, rod-shaped nanoparticles have demonstrated superior antibiofilm activity compared to spherical ones [66,67].

## 3. Materials and Methods

### 3.1. Sampling of Algae, Sponge, and Marine Sediment

Different samples of algae, marine sponge, and marine sediment were collected at depths ranging from 2 m to 15 m, in two Algerian littoral regions: Ain El-Turck (15 km from the north-west of Oran Province, Algeria) and El-Kala (20 km from the northeast of El Tarf Province, Algeria). The algae was identified as *Codium bursa*, while the marine sponge was identified as *Chondrosia reniformis* (Appendix A).

### 3.2. Isolation of Actinomycetes from the Marine Source

Prior to isolating actinobacteria, marine sediment samples were initially air-dried at room temperature. Subsequently, 1 g of sediment sample was diluted in 9 mL of distilled water, and serial dilutions were conducted from this stock solution until reaching 10^−3^. However, the marine sponge *Chondrosia reniformis* and the algae *Codium bursa*, first underwent an initial surface decontamination using 70% alcohol to eliminate loosely attached microorganisms. Afterward, the samples were cut into small pieces of 1 cm^3^ using a sterile scalpel and then aseptically ground using sterile pestles and mortars. In total, 10 g of each sample was suspended, separately, in 10 mL of sterile distilled water, from which a serial dilution was prepared. Aliquots (0.1 mL) of each dilution, prepared from different samples, were spread in duplicate on the surface of four culture media: ISP2: glucose 4 g/L, yeast extract 4 g/L, malt extract 10 g/L, agar 12 g, seawater 1 L, pH 7.2 ± 0.2. Casein starch agar (CSA): starch 10 g/L, casein 0.3 g/L, K_2_HPO_4_ 2 g/L, KNO_3_ 2 g/L, NaCl 2 g/L, MgSO_4_·7H_2_O 0.05 g/L, CaCO_3_ 0.02 g/L, FeSO_4_ ·7H_2_O 0.01 g/L, agar 15 g/L, seawater 1 L, pH 7.2 ± 0.2. M4: chitin 2 g/L, agar 18 g/L, seawater 1 L, pH 7.2 ± 0.2. M2: glycerol 6 mL, arginine 1 g/L, K_2_HPO_4_ 1 g/L, MgSO_4_ 0.5 g/L, agar 18 g/L, seawater 1 L, pH 7.2 ± 0.2. To prevent fungal growth, 50 μg/mL of cycloheximide was added to the four used media. Actinomycetes colonies were picked based on their morphological characteristics and then purified to achieve pure cultures.

### 3.3. Antimicrobial Activity

#### 3.3.1. Preparation of Suspensions

A total of seven bacteria were employed as test microorganisms, including four Gram-positive bacteria: *Staphylococcus aureus* (ATCC 25923), *Bacillus cereus* (ATCC 10876), *Listeria monocytogenes* (ATCC 35152), and *Micrococcus luteus* (ATCC), and three Gram-negative bacteria: *Pseudomonas aeruginosa* (ATCC 19606), *Escherichia coli* (ATCC 25922), and *Klebsiella pneumoniae* (ATCC 70603), in addition to one yeast: *Candida albicans* (ATCC 10231). The tested bacteria and yeast were inoculated into a brain–heart infusion broth and Sabouraud broth, respectively. After 24-h incubation, the turbidity of each culture was adjusted to 0.5 McFarland, after which, the cultures were inoculated onto Müller Hinton agar plates using sterile cotton swabs.

#### 3.3.2. Screening of Antimicrobial Activity Using the Agar Piece Method

To evaluate the antimicrobial potential of actinobacteria isolates, the strains were inoculated onto 5333 media: starch 15 g/L, yeast extract 4 g/L, MgSO_4_ 0.5 g/L, K_2_HPO_4_ 1 g/L, agar 18 g/L, seawater 1 L. After 14 days of incubation at 30 °C, agar cylinders of 6 mm diameter, were cut from the well-grown cultures of actinobacteria isolates and then placed onto pre-seeded Müller Hinton agar plates, containing the target microorganisms. The plates were placed at 4 °C for 4 h to ensure a good diffusion of bioactive metabolites secreted by actinomycete strains, followed by incubation at 37 °C for 24 h for the tested bacteria, and 30 °C for 48 h for the tested yeast [26].

### 3.4. Molecular Identification and Partial Taxonomic Characterization of Selected Strain

#### 3.4.1. Molecular Identification

The actinomycete isolates demonstrating significant antimicrobial activity were selected for molecular identification through 16S rRNA gene region sequencing. The DNA extraction was performed after the culture of actinomycetes isolates in ISP2 medium: malt extract 10 g/L, yeast extract 4.0 g/L, glucose 4.0 g/L, agar 12 g/L, seawater 1 L). The cultures were incubated at 30 °C for 5 days and 150 rpm. Subsequently, the biomass was harvested with centrifugation at 6000 rpm/min, and then DNA extraction was carried out from the biomass using a Spin Plant Mini Kit (Invisorb, Berlin, Germany). However, the amplification of the 16S rRNA gene region was conducted with PCR using two universal primers, such as 27F and R1492 [68]. The PCR reaction was conducted within microtubes of 200 µL, which contained 50 µL total reaction volume formed by: 22 μL of PCR water, 25 μL of JumpStar Ready Mix (JSRM, Sigma-Aldrich, Berlin, Germany), 1 μL forward primer, and 1 µL reverse primer, as well as 1 μL of template DNA. One tube, without DNA, was used as a control. The PCR reaction was performed in a Mastercycler Gradient (Eppendorf, Germany). The PCR products were purified with NucleoSpin^®^ Gel and PCR Clean-up-Kit (Macherey Nagel, Düren, Germany), and then recovered in 30 μL of elution buffer. The success of PCR was verified through 0.8% agarose gel electrophoresis for 50 min at 70 V in TAE buffer. The partial sequencing of the 16S rRNA gene of actinomycetes strains was accomplished with two primers, F27 and R518. The quality of the resulting sequences was checked manually using the Bioedit alignment program. Furthermore, the obtained sequences were aligned and compared with those available on the GenBank database (http://www.ncbi.nlm.nih.gov/BLAST, accessed on 10 March 2023). The phylogenetic tree was calculated with the software MEGA 7, using the neighbor-joining method [69,70]. The resulting tree topologies were evaluated using the bootstrap resampling method of 1000 replicates [71].

#### 3.4.2. Partial Characterization of Selected Strain

Partial taxonomic characterization was conducted on the most potent strain. This involved morphological, physiological, and biochemical characterization following the methods described by [72,73].

### 3.5. Silver Carbonate Nanoparticle Biosynthesis, Characterization, and Biological Activity Evaluation

#### 3.5.1. Biosynthesis of Silver Carbonate Nanoparticles

One piece of 1 cm^3^ was cut from a 14-day-old culture of the selected strain and then inoculated into an Erlenmeyer flask (250 mL) containing 100 mL of ISP2 medium. After 5 days of incubation at 30 °C in a shaker (120 rpm), the biomass was separated from the supernatant by centrifugation at 5000 rpm for 30 min. The collected biomass was subsequently subjected to two to three washes to eliminate the residual medium components adhered to the cells [74]. A 2 g portion from the obtained biomass of the selected strain, was suspended in a 250 mL Erlenmeyer flask containing 50 mL of silver nitrate (AgNO_3_) and 50 mL of calcium carbonate (CaCO_3_) solutions. Both solutions were prepared with an equal concentration of 500 mM in distilled water. Following a 72-h incubation period in the dark at 30 °C and under agitation (150 rpm), the biosynthesis of silver carbonate nanoparticles (BioAg_2_CO_3_NPs) was visually detected by observing a color shift from the initial pale yellow (the color of the biomass post-washing) to a dark brown.

The biosynthesized BioAg_2_CO_3_NPs were isolated from the biomass through ultrasonic disruption at 30 °C for 30 min, followed by high-speed centrifugation (18,000× *g* for 30 min). Subsequently, the collected BioAg_2_CO_3_NPs were dried at 100 °C [75].

#### 3.5.2. Characterization of Synthesized Silver Carbonate Nanoparticles

The characterization step of the synthesized BioAg_2_CO_3_NPs by the selected strain, aimed to gain a comprehensive understanding of their properties, composition, morphology, and size. To achieve this, the synthesized pellets of BioAg_2_CO_3_NPs were dispersed in sterile double-distilled water and centrifuged for 10 min at 10,000 rpm. The nanoparticle pellets were dried at 50 °C and were analyzed by X-ray diffraction (XRD) (Malvern Panalytical, Empyrean, Malvern, UK). To conduct elemental analysis of the BioAg_2_CO_3_NPs, biosynthesized by the selected *Actinobacteria* strain, the X-ray fluorescence (XRF) technique, was employed through a Thermo Scientific Niton™ XL3t 106399 XRF Analyzer (Billerica, MA, USA). The capping and stabilizing agent present in the surface of the BioAg_2_CO_3_NPs were analyzed by FTIR in the range of 400–4000 cm^−1^ at a resolution of 4 cm^−1^ using Tensor II, Bruker, (Billerica, MA, USA). The morphology of the biosynthesized nanoparticles was determined using scanning electron microscopy (ThermoFisher Scientific, Quattro, (Billerica, MA, USA), as well as the Atomic Force Microscope (AFM).

#### 3.5.3. Biological Activity Determination of Silver Carbonate Nanoparticles

##### Antimicrobial Activity

The agar well diffusion method was used to evaluate the antimicrobial activity of the BioAg_2_CO_3_NPs, synthesized using the biomass of the selected strain, against a panel of seven pathogenic microorganisms. However, the serial dilution method in 96-well microtiter plates was applied to determine the minimum inhibitory concentration (MIC) of BioAg_2_CO_3_NPs, against the tested microorganisms [74].

After the adjustment of microbial culture turbidity for each tested bacteria and yeast to 0.5 McFarland, the microbial suspension was inoculated onto the surfaces of Muller Hinton and Sabouraud medium, for bacteria and yeast, respectively. Afterward, wells with a diameter of 6 mm, were created in each Petri dish using a sterile cork borer. Subsequently, 20 µL of silver carbonate nanoparticle solution, prepared at a concentration of 150 µg/mL, was introduced into each well. The plates were incubated for 24 h at 30–37 °C, following the incubation of plates, the inhibition areas were measured [76].

The minimum inhibitory concentration (MIC) of BioAg_2_CO_3_NPs was determined using the serial dilution method in 96-well microtiter plates. The BioAg_2_CO_3_NPs synthesized using the biomass of the selected strain were tested against pathogenic microorganisms at final concentrations ranging from 150 μg/mL to 1.17 μg/mL, with distilled water used as the negative control. After incubation, the minimum inhibitory concentration (MIC) was determined as the lowest concentration visibly inhibiting the growth of each tested microorganism [77].

##### Antibiofilm Activity

Polystyrene 96-well microtiter plates were used to test the efficiency of BioAg_2_CO_3_NPs, prepared from the biomass of the selected strain, to eradicate the 24-h biofilm formed by *Klebsiella pneumoniae* kp6 strain. The strain used in this experience was isolated from urinary catheter and endotracheal tubes from the services of Intensive Care, Urology, and Neurology, at the University Hospital of Tlemcen. This strain was a high biofilm producer on inert surfaces as described previously by [78].

Each well of the microtiter plate was filled with 200 μL suspensions of *Klebsiella pneumoniae* kp6, adjusted to a turbidity of 0.5 McFarland. After incubation for 24 h at 37 °C, the non-adherent cells were gently removed by washing the biofilms three times with distilled water [79]. Simultaneously, the suspension of BioAg_2_CO_3_NPs was prepared at a concentration equivalent to the MIC determined against the *Klebsiella pneumoniae* ATCC 70603 strain. Then, 200 µL of this suspension was added to each well of the microtiter plate. After 24 h of incubation at 37 °C, the wells were rinsed three times, using distilled water. Afterward, the biofilm was stained with 200 μL of 0.1% crystal violet for 15 min, washed with tap water to remove excess stain, and then allowed to dry.

To quantify microbial adhesion, the stain in the wells was diluted with 200 μL acetic acid in water (33%), and the absorbance was measured at 595 nm with a microplate reader. Percentages of reduction in biofilm structures were calculated using the formula [79]:Biofilm inhibition (%)=[(OD control−OD sample) / OD control]×100

## 4. Conclusions

The present study represents the first attempt at the green synthesis of low-cost and eco-friendly silver carbonate nanoparticles using the biomass of the marine strain *Saccharopolyspora erythraea* S26, isolated from the Algerian coastline. The synthesized nanoparticles primarily exhibited a triangular morphology, with sizes ranging from 90 to 110 nm. Additionally, the synthesized BioAg_2_CO_3_NPs displayed good antibacterial and antibiofilm activity. These findings suggest that BioAg_2_CO_3_NPs can serve as an alternative approach to overcome microbial resistance to various antimicrobial agents since microbial resistance toward metallic nanoparticles is extremely rare.

## Figures and Tables

**Figure 1 marinedrugs-21-00536-f001:**
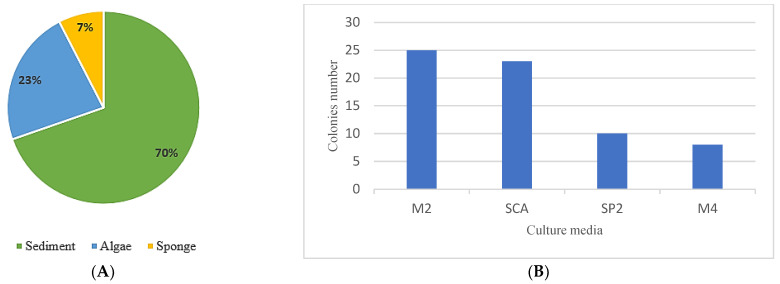
Distribution of the 66 *Actinobacteria* strains, according to (**A**) the percentage of actinomycete isolated from each marine source used, and (**B**) the number of actinomycete strain colonies obtained from each selective culture media used.

**Figure 2 marinedrugs-21-00536-f002:**
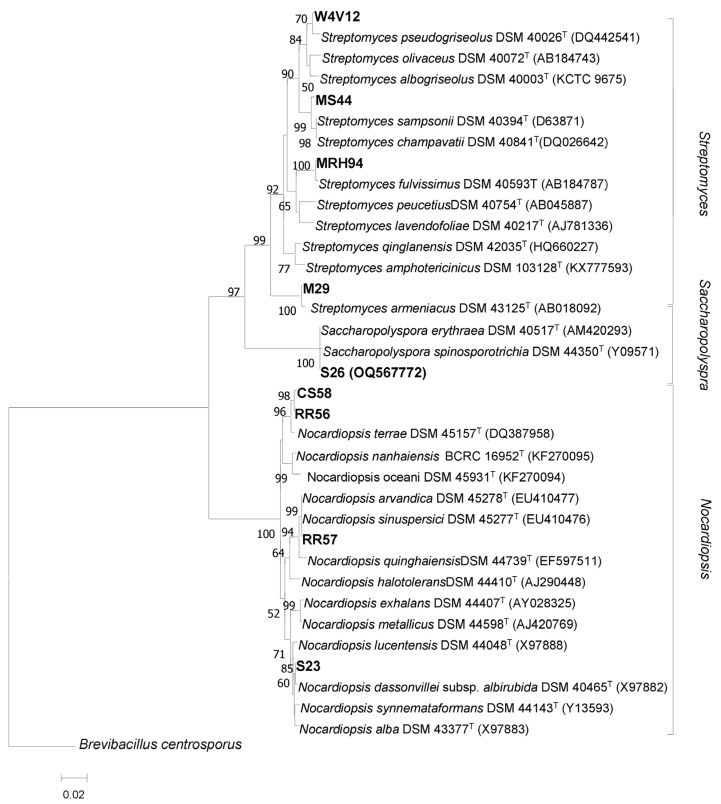
Neighbor-joining phylogenetic tree based on partial 16S rRNA gene sequencing, showing the phylogenetic relationship between the nine selected strains and the type strains of the closest species. Numbers at the nodes are bootstrap values, expressed as a percentage of 1000 resamplings (only values 50% are shown).

**Figure 3 marinedrugs-21-00536-f003:**
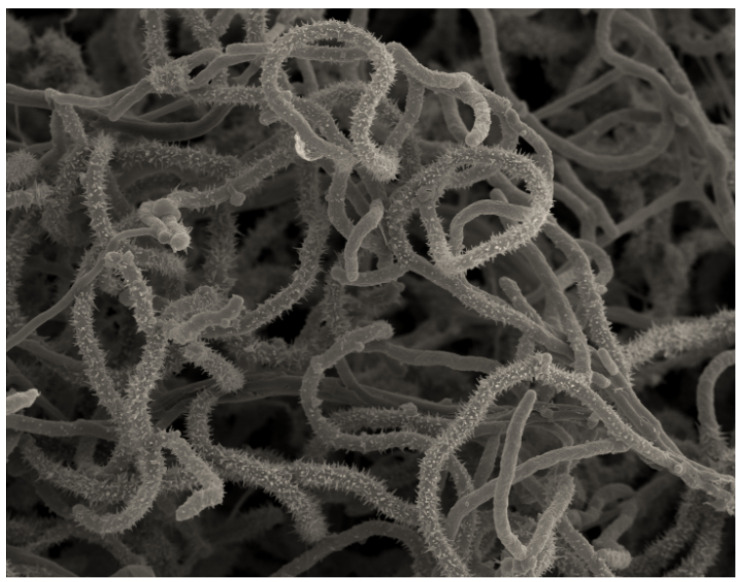
Micromorphology of aerial mycelium, seen by scanning electron microscopy, of isolate S26 showing spore chains surrounded by a spiny sheath. Scale: 200 nm.

**Figure 4 marinedrugs-21-00536-f004:**
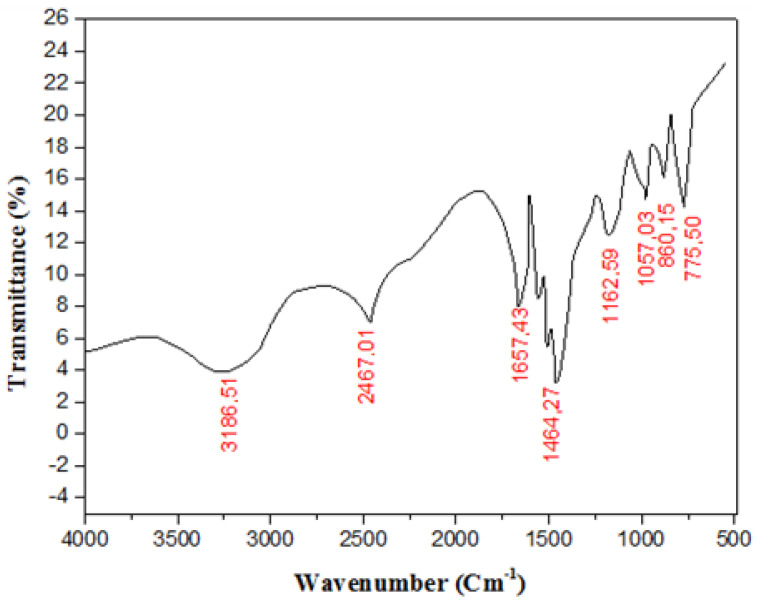
The FTIR spectrum of BioAg_2_CO_3_NPs synthesized using the biomass of *Saccharopolyspora erythraea* S26 strain.

**Figure 5 marinedrugs-21-00536-f005:**
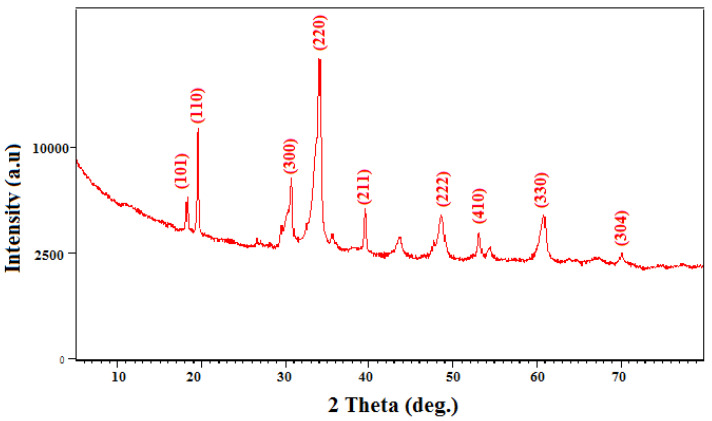
XRD pattern of silver carbonate nanoparticles (Ag_2_CO_3_NPs) synthesized using the biomass of strain *Saccharopolyspora erythraea* S26.

**Figure 6 marinedrugs-21-00536-f006:**
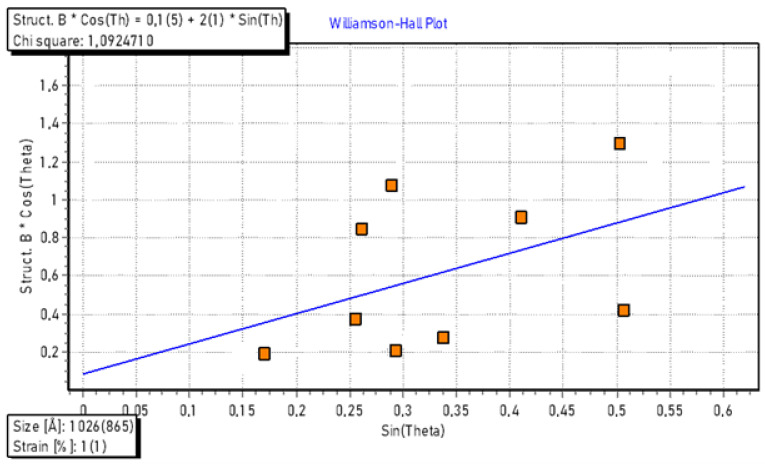
The Williamson–Hall plot of the BioAg_2_ CO_3_NPs nanoparticles.

**Figure 7 marinedrugs-21-00536-f007:**
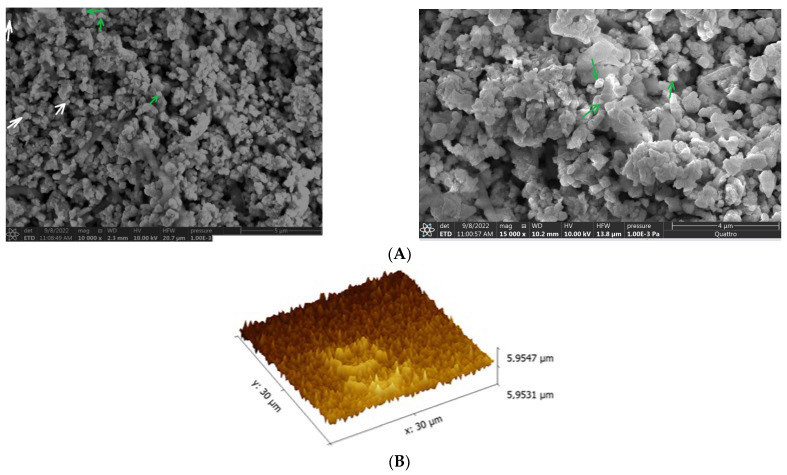
Scanning electron microscope (**A**) and atomic force microscopy (AFM) (**B**), of BioAg_2_CO_3_NPs synthesized from the biomass of strain *Saccharopolyspora erythraea* S26. Green arrow: BioAg_2_CO_3_NPs with triangular shape, white arrow: BioAg_2_CO_3_NPs with rectangular shape.

**Figure 8 marinedrugs-21-00536-f008:**
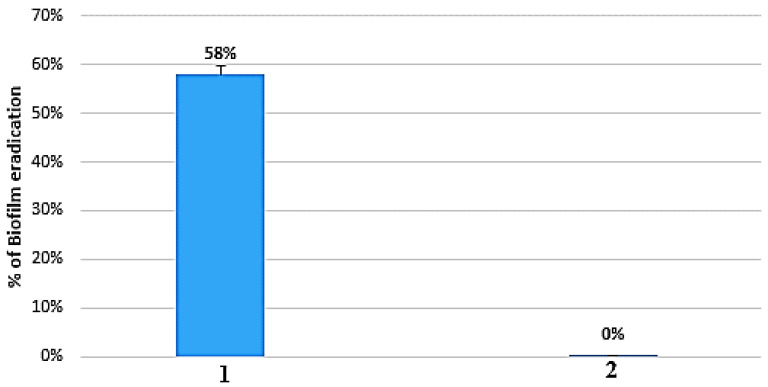
Antibiofilm activity of silver carbonate nanoparticles prepared from biomass of strain S26 against 24 h biofilm formed by *Klebsiella pneumoniae* kp6 strain. **1**: BioAg_2_CO_3_NPs; **2**: Control.

**Table 1 marinedrugs-21-00536-t001:** The atomic content of the BioAg_2_CO_3_NPs.

Element	ppm	+/−	±2σ
Mo	<LOD	:	7.792
Zr	<LOD	:	6.722
Sr	<LOD	:	5.530
U	<LOD	:	12.946
Rb	<LOD	:	6.935
Th	<LOD	:	10.550
Pb	<LOD	:	16.316
Au	<LOD	:	16.402
Se	<LOD	:	12.140
As	<LOD	:	12.232
Hg	<LOD	:	31.519
Zn	<LOD	:	24.793
W	<LOD	:	83.775
Cu	<LOD	:	50.723
Ni	<LOD	:	45.288
Co	<LOD	:	24.146
Fe	<LOD	:	91.363
Mn	<LOD	:	112.081
Ag	24,072.557	+/−	123.126
Ca	344.031	+/−	40.808
K	132.124	+/−	67.157
S	<LOD	:	250.576
Ba	<LOD	:	39.464
Cs	<LOD	:	9.806
Te	<LOD	:	25.027
Sb	<LOD	:	12.620
Sn	<LOD	:	63.174
Cd	<LOD	:	27.604

**Table 2 marinedrugs-21-00536-t002:** Antimicrobial activity of BioAg_2_CO_3_NPs tested against different microorganisms.

Tested Strains	*P. aeruginosa*	*M. luteus*	*B. subtilis*	*S. aureus*	*K. pneumoniae*	*E. coli*	*C. albicans*
	Inhibition zone diameters (mm)						
BioAg_2_CO_3_NPs	14	18.50	21	16	11	15	18
	**Minimum inhibitory concentration (µg/mL)**						
BioAg_2_CO_3_NPs	37.5	18.75	4.68	37.5	150	37.5	18.75
Oxytetracycline	1.66	2.08	2.08	0.52	1.66	3.33	n.a
Nystatine	n.a	n.a	n.a	n.a	n.a	n.a	

Positive control: oxytetracycline, nystatin; n.a: No activity.

## Data Availability

Not applicable.

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
