# Peer review of "Characterization of Silver Carbonate Nanoparticles Biosynthesized Using Marine Actinobacteria and Exploring of Their Antimicrobial and Antibiofilm Activity"

_marinedrugs, 2023, doi:10.3390/md21100536_

Round 1

Reviewer 1 Report

Comments for marinedrugs-2648403

      1. Line 46, Algerian coastline should be removed.

2. Line 79-80, in response to the lack of published research material on such an important topic is absolutely.

3. In figure 1A, word display is incomplete, 46%, 15% and 5% were added, but the results is not 100%.

4. Actinobacteriastrains in ine 92 and “whichincludes” in line 191 should be changed to two words.

5. Line 98-100, the sentence could be more concise.

6. Line 106, Archaea should be standardized form.

7. Line 116, Fucus sp” is wrong.

8. The number of different cultures media is different, 5 in line 96, 4 in line 124 and in figure 1B.

9. In Supplementary Table S1, the form of “isolates” should be same.

10. Line 151, there is absence of a comma in A. baumannii S. aureus .

11. In part 2.2, there is absence of discussion on the reason of the resistance of  L. monocytogenes to tested strains.

12. Line 167, “ (Figure 2)” should be removed.

13. Lien 175-178, , in “99,30 %”, “99,75%” and “ 99,70 %” should be ..

14. In Supplementary Table S2, the content in first line should be in English.

15. Why it was selected partial 16S rRNA gene sequence but not whole 6S rRNA gene sequence for identification of Actinobacteria strains?

16. In Supplementary figure S2, there were absence of “A”.

17. Line 266, “Table 2” should be “Table 1”.

18. The first letter of words in the sentence of “Table S4. Peak List of Silver nanoparticles biosynthesis using the biomass strain S26.”

19. The number in Ag2CO3 in Line 293, 322, 329, 537, 551 and 553 should be subscripts.

20. “ CO32−” in Line 306 should be agreement with “CO32− in line 303.

21. Why L. monocytogenes was absence in Antimicrobial activity part?

22. Line 340, the sentence should be removed.

23. line 376. “ BioAg2CO3NPsUV-vis” should be  BioAg2CO3NPs UV-vis.

24. Line 380, “Strain” should be strain”.

25. Line 389, “ BioAg2CO3NPscan” should be “ BioAg2CO3NPs can”.

26. Line 391-392, “ BioAg2CO3NPsmay should be “ BioAg2CO3NPs may.

27. “alga” in line 404 and 410 should be “algae.

28. “ Actinobacteria” in line 407, 437 and 440 should be actinobacteria”.

29. Line 410, “Chondrosiareniformisand” should be “ Chondrosiareniformis and”.

30. Line 418, “MgSO4.7H2O” should be “MgSO4·7H2O”.

31. Line 419, “FeSO4 .7H2O” should be “FeSO4 ·7H2O”.

32. The number in MgSO4 and K2HPO4 in line 438, AgNO3 in line 483, and CaCO3 In line 484 should be subscripts.

33. Line 486, “ 30°C” should be “ 30 °C”. Line 540, “37°C” should be “37 °C” .

34. Line 509, six should be seven.

35. Line 519, “ Following” should be following”.

36. Line 523, the unit in 150 µg/ml to 1.17 µl/ml is not agreement.

37. Line 550, “ Saccharopolysporaerytherea” should be italic.

38. Is there any necessary relationship between the antibacterial activity of actinomycetes and the preparation of nano silver? If not, it is not meaningful to determine the antibacterial activity of isolated actinomycetes. It need simply screen for actinomycetes that can produce silver nanoparticles.

39. Line 321-322, “Moreover, the SEM results, indicate that the average size of BioAg2CO3NPs ranges from 90 nm to 110 nm”. But in part of Conclusions, size ranges from 26 to 30 nm. They are inconsistent.

40. Line 318-321, “The prevalent form observed was triangular (indicated by the green arrow) (Figure 7. A), while other shapes, such as rectangular (indicated by the white arrow) and spherical, were also detected (Figure 7. A).” but in part of Conclusions, “ triangular silver carbonate nanoparticles”, they are inconsistent.

Minor editing of English language required.

Author Response

Dear Editor and Reviewers,

Thank you for your letter and for the reviewers’ comments concerning our manuscript entitled “Characterization of Silver Carbonate Nanoparticles Biosynthesized using Marine Actinobacteria and Exploring of their Antimicrobial and Antibiofilm Activity”. Those comments are all really valuable and helpful in revising and improving our manuscript, as well as providing significant direction for our research. We carefully considered the comments and made changes that we hope will be approved. The revised sections of the document are highlighted in yellow. The following are the main corrections in the manuscript and responses to the reviewers’ comments.

Reviewer #1:

  1. Line 46, “Algerian coastline” should be removed.

Author’s response: Thank you for your comments; "The term 'Algerian coastline' has been removed." (Please see the revised manuscript)

  1. Line 79-80, “in response to the lack of published research material on such an important topic” is absolutely

Author’s response: Thank you for your valuable comments;

  1.  In figure 1A, word display is incomplete, 46%, 15% and 5% were added, but the results is not 100%.

Author’s response: Thank you for your valuable comments; Figure 1A has been corrected, and now the sum of the results equals 100%. (Please see the revised manuscript)

  1. “Actinobacteriastrains” in ine 92 and “whichincludes” in line 191 should be changed to two words.

Author’s response: Thank you for your comments; "The "Actinobacteria strains" and "which includes" have been corrected as two separate words: 'Actinobacteria strains' and 'which includes'."

  1. Line 98-100, the sentence could be more concise.

Author’s response: Thank you for your comments; The sentence has been made concise:

46 strains were from marine sediment (70%), 15 isolates from the marine algae Codium bursa (23 %), and 5 strains from the marine sponge Chondrosia reniformis (7 %) (Figure 1, A). (Please see the revised manuscript)

  1. Line 106, “Archaea” should be standardized form.

Author’s response: Thank you for your comments; “Archaea” has been standardized to:Previous studies have reported that sponges and algae host substantial microbial biomass, encompassing archaea, fungi, viruses, and bacteria

  1. Line 116, “Fucussp” is wrong.

Author’s response: Thank you for your comments;

'Fugus sp' has been deleted,

Fugus sp” Has been deleted and the reference: Djinni, I. et al. Streptomyces thermoviolaceus SRC3 strain as a novel source of the antibiotic adjuvant streptazolin: A statistical approach toward the optimized production. Journal of microbiological methods 148 (2018) 161-168. https://doi.org/10.1016/j.mimet.2018.04.008

The reference above has been replaced with the correct one."

Djinni, I.; Defant, A.; Kecha, M.; Mancini, I. Metabolite profile of marine-derived endophytic Streptomyces sundarbansensis WR 1 L 1 S 8 by liquid chromatography–mass spectrometry and evaluation of culture conditions on antibacterial activity and mycelial growth. J. Appl. Microbiol. 2014, 116, 39–50. https://doi.org/10.1111/jam.12360

(Please see the revised manuscript)

  1. The number of different cultures media is different, 5 in line 96, 4 in line 124 and in figure 1B.

Author’s response: Thank you for your comments;

The correct culture media number was 4; therefore, “five” in line 96 has been corrected to:

….. using four different culture media…..  (Please see the revised manuscript)

  1. In Supplementary Table S1, the form of “isolates” should be same.

Author’s response: Thank you for your comments;

The word 'Isolats' in the 'Supplementary Table S1' has been corrected to 'isolates'. (Please see the revised manuscript)

  1. Line 151, there is absence of a comma in “A. baumannii S. aureus” .

Author’s response: Thank you for your comments;

A comma between A. baumannii ,and S. aureus has been added (Please see the revised manuscript)

  1. In part 2.2, there is absence of discussion on the reason of the resistance of  L. monocytogenesto tested strains.

Author’s response: Thank you for your comments;

The discussion about the reasons for the resistance of L. monocytogenes to the secondary metabolites secreted by actinomycetes strains has been included:

Regarding resistance, L. monocytogenes was the most resistant bacteria tested. This resistance can be attributed to intrinsic mechanisms, such as efflux pumps, which can pump out or neutralize antimicrobial compounds, making the eradication of L. monocytogenes challenging

  1. Line 167, “ (Figure 2)” should be removed.

Author’s response: Thank you for your comments (Figure 2) has been deleted.   (Please see the revised manuscript)

  1. Line 175-178, “,” in “99,30 %”, “99,75%” and “ 99,70 %” should be “.”.

Author’s response: Thank you for your comments "Commas have been replaced with "." in '99.30%', '99.75%', and '99.70%'."

  1. In Supplementary Table S2, the content in first line should be in English.

Author’s response: Thank you for your comments The content in Supplementary Table S2 has been corrected and is now in English.

  1. Why it was selected partial 16S rRNA gene sequence but not whole 6S rRNA gene sequence for identification of Actinobacteriastrains?

Author’s response: Thank you for your comments In Actinobacteria identification based on 16S rRNA gene sequencing, we typically begin by using two primers, namely 27F and R518. The obtained sequences are then assembled to generate a single partial sequence ranging from 800 to 900 base pairs (bp) in length. This sequence is subsequently compared to sequences stored in the public database (NCBI). If the similarity between the identified strain and the closest species falls below the cutoff value, which is set at 98.6% for distinguishing between two bacterial species, or is only slightly higher, for instance, below 90%, we can perform additional sequencing using primers such as F1100, R1100, and R1525 to obtain the full 16S rDNA sequence. However, in the case of strain S26, which exhibited a 99.70% similarity to the species Saccharopolyspora erythraea, there is no need to use additional primers. This is because strain S26 has been identified as a known species, a conclusion that was confirmed using a polyphasic approach.

  1. In Supplementary figure S2, there were absence of “A”.

Author’s response: Thank you for your comments “A” has been added to Supplementary Figure S1.

  1. Colony description and melanin pigment formation; from top left to bottom right: ISP2, ISP3, ISP4, ISP5, ISP6 and ISP7
  2. Line 266, “Table 2” should be “Table 1”.

Author’s response: Thank you for your comments 'Table 1' has been corrected to 'Table 2'.

  1. The first letter of words in the sentence of “Table S4. Peak List of Silver nanoparticles biosynthesis using the biomass strain S26.”

Author’s response: Thank you for your comments "The first letters of the words in the sentence of Table S4 have been changed to lowercase."

Table S4. Peak list of silver carbonate nanoparticles biosynthesis using the biomass strain S26.

  1. The number in Ag2CO3 in Line 293, 322, 329, 537, 551 and 553 should be subscripts.

Author’s response: Thank you for your comments The numbers 2 and 3 in 'Ag2CO3' have been used as subscripts throughout the entire manuscript.

  1. “ CO32−” in Line 306 should be agreement with “CO32−” in line 303.

 Author’s response: Thank you for your comments.

"The 'CO32−' has been corrected to “CO32−” 

  1. Why L. monocytogenes was absence in Antimicrobial activity part?

Author’s response: Thank you for your comments I agree with your suggestion that testing BioAg2CO3NPs against L. monocytogenes would be valuable. Unfortunately, in the current study, we did not include L. monocytogenes in the antimicrobial testing of BioAg2CO3NPs.

22.Line 340, the sentence should be removed.

Author’s response: Thank you for your comments "The sentence has been deleted.

  1. line 376. “ BioAg2CO3NPsUV-vis” should be “ BioAg2CO3NPs UV-vis”.

Author’s response: Thank you for your comments “ BioAg2CO3NPsUV-vis” has been corrected to “ BioAg2CO3NPs UV-vis”.

  1. Line 380, “Strain” should be “strain”.

Author’s response: Thank you for your comments 'Strain' has been corrected to 'strain'.

25.Line 389, “ BioAg2CO3NPscan” should be “ BioAg2CO3NPs can”.

Author’s response: Thank you for your comments “ BioAg2CO3NPscan” has been corrected “ BioAg2CO3NPs can”.

  1. Line 391-392, “ BioAg2CO3NPsmay” should be “ BioAg2CO3NPs may”.

Author’s response: Thank you for your comments “ BioAg2CO3NPsmay” has been corrected to “ BioAg2CO3NPs may”.

  1. “alga” in line 404 and 410 should be “algae”.

Author’s response: Thank you for your comments Alga has been corrected to “algae”

28.“ Actinobacteria” in line 407, 437 and 440 should be “actinobacteria”.

Author’s response: Thank you for your comments Actinobacteria” has been corrected to “actinobacteria

  1. Line 410, “Chondrosiareniformisand” should be “ Chondrosiareniformis and”.

Author’s response: Thank you for your comments “Chondrosiareniformisand” has been corrected to “ Chondrosiareniformis and

  1. 30. Line 418, “MgSO4.7H2O” should be “MgSO47H2O”.

Author’s response: Thank you for your comments “MgSO4.7H2O” has been corrected to “MgSO4·7H2O”.

  1. 31. Line 419, “FeSO4.7H2O” should be “FeSO47H2O”.

Author’s response: Thank you for your comments “FeSO4 .7H2O” has been corrected to “FeSO4 ·7H2O”.

32.The number in MgSO4 and K2HPO4 in line 438, AgNO3 in line 483, and CaCO3 In line 484 should be subscripts.

Author’s response: Thank you for your comments The numbers in MgSO4, K2HPO4, AgNO3 and CaCO3, has been subscripts.

  1. Line 486, “ 30°C” should be “ 30 °C”. Line 540, “37°C” should be “37 °C” .

Author’s response: Thank you for your comments "30°C" and "37°C" have been corrected to "30 °C" and "37 °C," respectively, in the whole manuscript.

  1. Line 509, “six” should be “seven”.

Author’s response: Thank you for your comments Six has been corrected to seven

  1. Line 519, “ Following” should be “ following”.

Author’s response: Thank you for your comments “Following” has been corrected to “ following”.

  1. Line 523, the unit in “150 µg/ml to 1.17 µl/ml” is not agreement.

Author’s response: Thank you for your comments The unit in 1.17 µl/ml” has been corrected to 1.17 µg/ml” 

  1. Line 550, “ Saccharopolysporaerytherea” should be italic.

Author’s response: Thank you for your comments Saccharopolysporaerytherea has been corrected to Saccharopolyspora erytherea

  1. 38. Is there any necessary relationship between the antibacterial activity of actinomycetes and the preparation of nano silver? If not, it is not meaningful to determine the antibacterial activity of isolated actinomycetes. It need simply screen for actinomycetes that can produce silver nanoparticles.

Author’s response: Thank you for your comments There is actually a direct relationship between the antimicrobial activity of the different actinobacteria isolates and the biosynthesis of nanoparticles. General strains with good antimicrobial potential can be good candidates for the biosynthesis of metallic nanoparticles, as they are able to produce secondary metabolites capable of reducing metallic ions to form metallic nanoparticles. For this reason, we have selected the strain S26 because it was the most active strain among the 66 actinobacteria strains obtained.

  1. Line 321-322, “Moreover, the SEM results, indicate that the average size of BioAg2CO3NPs ranges from 90 nm to 110 nm”. But in part of Conclusions, “size ranges from 26 to 30 nm”. They are inconsistent.

Author’s response: Thank you for your comments "The correct average size of BioAg2CO3NPs ranges from 90 nm to 110 nm. The sentence in the conclusion has been corrected to

……with size ranges from 90 to 110 nm, using the biomass of the marine strain Saccharopolyspora erytherea S26

  1. Line 318-321, “The prevalent form observed was triangular (indicated by the green arrow) (Figure 7. A), while other shapes, such as rectangular (indicated by the white arrow) and spherical, were also detected (Figure 7. A).” but in part of Conclusions, “ triangular silver carbonate nanoparticles”, they are inconsistent.

Author’s response: Thank you for your comments

I agree with you, and the sentence in the conclusion has been corrected to:

The synthesized nanoparticles primarily exhibit a triangular morphology, with sizes ranging from 90 to 110 nm (Please see the revised manuscript)

The language of the manuscript has been revised and updated

Reviewer 2 Report

The article titled Characterization of Silver Carbonate Nanoparticles Biosynthesized using Marine Actinobacteria and Exploring of their Anti-microbial and Antibiofilm Activity is accepted after consideration of the following minor comments.

1)    Abstract, the results of most active compound and reference standard should be added for comparison .

2)    Saccharopolysporaerythrea was known to have significant antimicrobial activity so what is the aim of your work.

3)    Authors mentioned in the introduction resistance of microorganism but I think they concerned of bacteria not all microorganism so they should concern on bacteria.

4)    The rational of study should be improve.

5)    Also standard must be used for comparing your compounds with it

Author Response

Dear Editor and Reviewers,

Thank you for your letter and for the reviewers’ comments concerning our manuscript entitled “Characterization of Silver Carbonate Nanoparticles Biosynthesized using Marine Actinobacteria and Exploring of their Antimicrobial and Antibiofilm Activity”. Those comments are all really valuable and helpful in revising and improving our manuscript, as well as providing significant direction for our research. We carefully considered the comments and made changes that we hope will be approved. The revised sections of the document are highlighted in yellow. The following are the main corrections in the manuscript and responses to the reviewers’ comments.

Reviewer #2:

Comments and Suggestions for Authors

The article titled Characterization of Silver Carbonate Nanoparticles Biosynthesized using Marine Actinobacteria and Exploring of their Anti-microbial and Antibiofilm Activity is accepted after consideration of the following minor comments.

  1. Abstract, the results of most active compound and reference standard should be added for comparison.

Author’s response: Thank you for your comments; " For comparison, the results of the most active compound and reference standard have been included." (Please see the revised manuscript)

For the assessment of the antimicrobial activity of BioAg2CO3NPs against various pathogenic microorganisms, two positive controls, oxytetracycline (for bacteria) and nystatin (for fungi), were employed. The results are presented in Table 2.

  1. Saccharopolysporaerythreawas known to have significant antimicrobial activity so what is the aim of your work.

Author’s response: Thank you for your valuable comments; Actinobacteria, such as Saccharopolyspora erythraea, are renowned for their capability to synthesize a wide array of secondary metabolites. Actinobacteria with strong antimicrobial activity can be excellent candidates for the biosynthesis of metallic nanoparticles. This is due to their ability to produce secondary metabolites capable of reducing metallic ions, which facilitate the formation of metallic nanoparticles. Additionally, they secrete biomolecules that can serve as capping or stabilizing agents, effectively preventing nanoparticle aggregation. Therefore, the strain S26, identified as belonging to the species Saccharopolyspora erythraea, has been selected for the biosynthesis of nanoparticles because it was the most active strain among the 66 actinobacterial strains obtained.

In addition, this study aims to biosynthesize silver carbonate nanoparticles using a simple method. It is important to note that the synthesis of Ag2CO3NPs has previously been reported using chemical or physical methods. However, this study was the first to employ a green method for synthesizing this type of nanoparticle."

  1. Authors mentioned in the introduction resistance of microorganism but I think they concerned of bacteria not all microorganism so they should concern on bacteria.

Author’s response: Thank you for your valuable comments; I agree with you, and the introduction has been corrected to focus on bacterial resistance rather than all microorganisms.

Bacterial resistance to different antimicrobial agents, such as antibiotics, disinfectants, antiseptics, and food preservatives, arises when microbial cells become no longer sensitive to the antimicrobial agents at the concentrations used in practice. This resistance has significant health, economic, and environmental implications [1]. Compared to planktonic cells, various studies have demonstrated that bacterial resistance increases significantly when cells are in the biofilm state. Biofilm represents a unique microbial lifestyle, where bacteria are attached to a solid surface and enclosed within a self-produced extracellular polymeric substance, primarily composed of exopolysaccharides and proteins [2,3]. Consequently, both natural and synthetic classes of antimicrobial agents are currently unable to evade this phenomenon [4]. The current progress in addressing both bacterial resistance and biofilm problems involves the application of metallic nanoparticles obtained from green synthesis

  1. The rational of study should be improve.

Author’s response: Thank you for your valuable comments. The rationale has been improved

  1. Also standard must be used for comparing your compounds with it

Author’s response: Thank you for your valuable comments The standard used has been presented in Table 2.

(Please see the revised manuscript)

Reviewer 3 Report

The authors identified an actinomycete strains S26, identified as Saccharopolysporaerythrea and synthesized silver carbonate nanoparticles (BioAg2CO3NPs) with S26. The results showed that BioAg2CO3NPs were in triangular shape with an approximate size of 100 nm and exhibited good antimicrobial and biofilm removing activity. The preparation of BioAg2CO3NPs is very simple and the anti-biofilm activity of BioAg2CO3NPs is very high. The research is meaningful but the manuscript needs to be revised thoroughly for the gramma mistakes. There are too many commas in some sentence and tense errors should be corrected.  

A total of 66 Actinobacteria strains, were isolated from different marine sources, and 95 collected from two sites along the Algerian littoral, using five different culture media. The 96 obtained Actinobacteria strains were distributed based on their marine origins as follows: 97 46 actinomycete strains (46%) were sourced from marine sediment, 15 isolates (15%) were 98 retrieved from the marine algae Codium bursa, and 5 actinomycete strains were obtained 99 from the marine sponge Chondrosia reniformis (Figure 1, A).

There are too many commas in the first sentence. The first two commas should be removed.

In Figure 1. B, there are four culture media. However, in the text of Line 96, there authors used five different culture media.

In Line 98, why 46 46 actinomycete strains were sourced from marine sediment? A total of 66 Actinobacteria strains plus 46% should be 30? This sentence is confusing.

In Line 652, the reference 36 is in wrong style.

In Line 118, please check the name of the compound.

In line 122-123, please check the grammar the sentence. Please pay attention to the tense consistency.

In line 132, the tense in these two sentences should be the same. There are too many grammar mistakes.

Why some of the number of the references in the manuscript were in bold type, and some were not?

In line 321 and 329, the number in BioAg2CO3NPs should be subscript.

In line 347, the unit for the MIC values is different from unit in Table2.

In line 340, there abbreviations of AgNPs mas and AgNPs Sup were not used in Table2. Why there are comments for them?  

In line 376, there is a lack of comma after BioAg2CO3NPs.

It is contradictory between Line 560 and Line 563.

Need extensive editing of English language.

Author Response

Dear Editor and Reviewers,

Thank you for your letter and for the reviewers’ comments concerning our manuscript entitled “Characterization of Silver Carbonate Nanoparticles Biosynthesized using Marine Actinobacteria and Exploring of their Antimicrobial and Antibiofilm Activity”. Those comments are all really valuable and helpful in revising and improving our manuscript, as well as providing significant direction for our research. We carefully considered the comments and made changes that we hope will be approved. The revised sections of the document are highlighted in yellow. The following are the main corrections in the manuscript and responses to the reviewers’ comments.

Reviewer #3:

The authors identified an actinomycete strains S26, identified as Saccharopolysporaerythrea and synthesized silver carbonate nanoparticles (BioAg2CO3NPs) with S26. The results showed that BioAg2CO3NPs were in triangular shape with an approximate size of 100 nm and exhibited good antimicrobial and biofilm removing activity. The preparation of BioAg2CO3NPs is very simple and the anti-biofilm activity of BioAg2CO3NPs is very high. The research is meaningful but the manuscript needs to be revised thoroughly for the gramma mistakes. There are too many commas in some sentence and tense errors should be corrected.  

  1. A total of 66 Actinobacteria strains, were isolated from different marine sources, and 95 collected from two sites along the Algerian littoral, using five different culture media. The 96 obtained Actinobacteria strains were distributed based on their marine origins as follows: 97 46 actinomycete strains (46%) were sourced from marine sediment, 15 isolates (15%) were 98 retrieved from the marine algae Codium bursa, and 5 actinomycete strains were obtained 99 from the marine sponge Chondrosia reniformis (Figure 1, A).  

There are too many commas in the first sentence. The first two commas should be removed.

Author’s response: Thank you for your comments; The first two commas from the first sentence have been removed, and the paragraph becomes:

A total of 66 Actinobacteria strains were isolated from different marine sources and collected from two sites along the Algerian littoral, using four different culture media. The obtained Actinobacteria strains were distributed based on their marine origins as follows: 46 strains were from marine sediment (70%), 15 isolates from the marine algae Codium bursa (23 %), and 5 strains from the marine sponge Chondrosia reniformis (7 %) (Figure 1, A).  (Please see the revised manuscript)

  1. In Figure 1. B, there are four culture media. However, in the text of Line 96, there authors used five different culture media.

Author’s response: We appreciate your valuable comments and apologize for the printing mistake; The word "five" in line 96 has been corrected to the number "four" because the actual culture media number was 4, not "five":

(Please see the revised manuscript)

  1. In Line 98, why 46 46 actinomycete strains were sourced from marine sediment? A total of 66 Actinobacteria strains plus 46% should be 30? This sentence is confusing.

Author’s response: I agree with you that the sentence was confusing. Actually, the total number of actinomycetes strains isolated from the three different marine sources was 66, distributed as follows:

  • 46 strains were from marine sediment, representing 70% of the total Actinobacteria strains obtained (out of 66 isolates of actinomycetes).
  • 15 isolates were from the marine algae Codium bursa, accounting for 23%.
  • 5 strains were from the marine sponge Chondrosia reniformis, comprising 7%.

The sentence and Figure 1.A have been corrected:

The obtained Actinobacteria strains (66 isolates) were distributed based on their marine origins as follows: 46 strains were from marine sediment (70%), 15 isolates from the marine algae Codium bursa (23 %), and 5 strains from the marine sponge Chondrosia reniformis (7 %) (Figure 1, A).

summarized in Figure 1.

A

B

Figure 1. Distribution of the 66 Actinobacteria strains, according to; A: The percentage of actinomycete isolated from each marine source used. B: The number of actinomycete strain colonies obtained from each selective culture media used.

(Please see the revised manuscript)

  1. In Line 652, the reference 36 is in wrong style.

Author’s response: Reference 36 has been corrected to :

Huang, Z.; Mo, S.; Yan, L.; Wei, X.; Huang, Y.; Zhang, L.; Zhang, S.; Liu, J.; Xiao, Q.; Lin, H.; et al. A simple culture method enhances the recovery of culturable actinobacteria from coastal sediments. Front. Microbiol. 202112, 1451. https://doi.org/10.3389/fmicb.2021.675048

(Please see the revised manuscript)

  1. In Line 118, please check the name of the compound.

Author’s response: Thank you for your comments The name of the compound has been checked, and it was found to be correct, taken directly from the publication of Djinni et al., 2013.

  1. In line 122-123, please check the grammar the sentence. Please pay attention to the tense consistency.

Author’s response: Thank you for your comments Both the grammar and the tense consistency have been corrected

Moreover, previous studies have shown that the composition of culture media significantly affects the number and diversity of Actinobacteria strains [26-28]. Therefore, to enhance the isolation of Actinobacteria from the three marine sources, four culture media were used.

  1. In line 132, the tense in these two sentences should be the same. There are too many grammar mistakes.

Author’s response: Thank you for your comments Both the grammar and the tense consistency have been corrected

Actinobacteria isolates were evaluated for their antibacterial activity against a panel of three Gram-negative bacteria and four Gram-positive bacteria, in addition to one yeast. The results are presented in Supplementary Table S1. The results reveal that 40 strains, out of 66 obtained isolates, exhibit antimicrobial activity against at least one of the tested microorganisms.

  1. Why some of the number of the references in the manuscript were in bold type, and some were not?

Author’s response: Thank you for your comments: Thank you for your comments: We apologize for the printing mistake; all references in the manuscript now have a uniform type. (Please see the revised manuscript)

  1. In line 321 and 329, the number in BioAg2CO3NPs should be subscript.

Author’s response: Thank you for your comments: The numbers in BioAg2CO3NPs have been subscripted in the whole manuscript

  1. In line 347, the unit for the MIC values is different from unit in Table2.

Author’s response: Thank you for your comments: The unit for the MIC values has been corrected to µg/mL.

 The results demonstrate that BioAg2CO3NPs are efficient antibacterial agents against all tested microbes, with MIC values ranging from 18.75 to 150 µg/mL. (Please see the revised manuscript)

  1. In line 340, there abbreviations of AgNPs mas and AgNPs Sup were not used in Table2. Why there are comments for them?  

Author’s response: Thank you for your comments: We apologize for the printing mistake; all references in the manuscript now have a uniform type. abbreviations, 'AgNPs mas' and 'AgNPs Sup,' have been deleted from the manuscript. (Please see the revised manuscript)

  1. In line 376, there is a lack of comma after BioAg2CO3

Author’s response: Thank you for your comments: A Comma has been added after BioAg2CO3NPs

In order to quantify the anti-biofilm activity of BioAg2CO3NPs, UV-vis spectroscopy was performed. The results are presented in Figure 8.

  1. It is contradictory between Line 560 and Line 563.

Author’s response: Thank you for your comments: The project number only indicates technical support; it does not represent funding.
